# Bidirectional dysregulation of synaptic glutamate signaling after transient metabolic failure

**Stefan Passlick[1], Ghanim Ullah[2], Christian Henneberger[3]***

[1]Institute of Cellular Neurosciences, Medical Faculty, University of Bonn, Bonn, Germany; [2]Department of Physics, University of South Florida, Tampa, United States; [3]German Center for Neurodegenerative Diseases (DZNE), Bonn, Germany

*For correspondence:
christian.henneberger@uni-bonn.de

Competing interest: The authors declare that no competing interests exist.

**Abstract** Ischemia leads to a severe dysregulation of glutamate homeostasis and excitotoxic cell damage in the brain. Shorter episodes of energy depletion, for instance during peri-infarct depolarizations, can also acutely perturb glutamate signaling. It is less clear if such episodes of metabolic failure also have persistent effects on glutamate signaling and how the relevant mechanisms such as glutamate release and uptake are differentially affected. We modeled acute and transient metabolic failure by using a chemical ischemia protocol and analyzed its effect on glutamatergic synaptic transmission and extracellular glutamate signals by electrophysiology and multiphoton imaging, respectively, in the mouse hippocampus. Our experiments uncover a duration-dependent bidirectional dysregulation of glutamate signaling. Whereas short chemical ischemia induces a lasting potentiation of presynaptic glutamate release and synaptic transmission, longer episodes result in a persistent postsynaptic failure of synaptic transmission. We also observed unexpected differences in the vulnerability of the investigated cellular mechanisms. Axonal action potential firing and glutamate uptake were surprisingly resilient compared to postsynaptic cells, which overall were most vulnerable to acute and transient metabolic stress. We conclude that short perturbations of energy supply lead to a lasting potentiation of synaptic glutamate release, which may increase glutamate excitotoxicity well beyond the metabolic incident.

## eLife assessment

The authors show that short bouts of chemical ischemia lead to presynaptic changes in glutamate release and long-term potentiation, whereas longer bouts of chemical ischemia lead to synaptic failure and presumably cell death. This **convincing** work relies on rigorous electrophysiology/ imaging experiments and data analysis. It is **important** as it provides new mechanistic details on chemical ischemia, which could offer potential insights into ischemic stroke in vivo.

## Introduction

It is well established that the breakdown of ATP production in the brain in an ischemic stroke profoundly affects glutamate signaling on various levels. The inability of neurons and other cell types such as astrocytes to maintain transmembrane ionic gradients and their membrane potential leads to increased glutamate release, decreased glutamate clearance, and, thereby, increased glutamate receptor activation and excitotoxicity, which ultimately results in cell death (*Dirnagl et al., 1999*). The ischemic core is also the source of peri-infarct depolarizations (PIDs) that travel into the primarily undamaged tissue. They are a relatively short-lasting subtype of spreading depolarizations associated with a significant metabolic burden that can lead to the progressive expansion of tissue damage in the

peri-infarct zone (*Lauritzen et al., 2011*). In animal models of ischemic stroke, they can be detected as transient glutamate increases lasting about a minute using for instance microdialysis and fluorescence imaging (*Lauritzen et al., 2011*; *Rakers and Petzold, 2017*). In addition to neuronal glutamate release, glial glutamate release and a transient reduction of glutamate uptake could contribute to such transient glutamate increases (*Lauritzen et al., 2011*; *Rakers and Petzold, 2017*; *Passlick et al., 2021*). Whether PIDs, and spreading depolarizations in general, acutely trigger lasting changes of glutamate signaling and homeostasis is far less clear.

Spreading depression is accompanied by a transient decrease of ATP levels (*Mies and Paschen, 1984*). This decrease can be reproduced in vitro by creating anoxic conditions, removal of substrates for ATP production such as glucose, inhibition of metabolic pathways, or combinations of these manipulations. In vitro studies using such approaches to induce a transient metabolic failure have demonstrated for instance that brief periods of anoxia and aglycemia lead to a persistent potentiation of synaptic transmission at CA3-CA1 synapses in the hippocampus, also called ischemic or anoxic long-term potentiation (LTP) (*Crépel et al., 1993*; *Crepel et al., 2003*; *Hsu and Huang, 1997*; *Stein et al., 2015*). This indicates that a transient metabolic failure can persistently alter glutamatergic synaptic transmission. Whether this also applies to glutamate release and glutamate uptake is currently not known but potentially important, because that may increase neuronal excitability and excitotoxicity in between individual PIDs and thereby tissue damage.

For these reasons, we explored if and how short and transient periods of metabolic failure affect glutamate release and uptake in parallel to synaptic transmission using combinations of electrophysiology and glutamate imaging in the hippocampus together with a chemical ischemia protocol. We observed a duration-dependent bidirectional dysregulation of glutamate release and synaptic transmission, in which short durations of chemical ischemia led to a persistent potentiation of glutamate release while long durations induced postsynaptic failure. We further uncovered unexpected differences of vulnerability of mechanisms involved in glutamatergic synaptic transmission, including a marked resilience of glutamate uptake to chemical ischemia.

## Results

### Bidirectional dysregulation of glutamate signaling

For revealing the effect of acute and transient metabolic failure on glutamatergic synaptic transmission, we combined electrophysiological monitoring of glutamatergic synaptic transmission at CA3-CA1 Schaffer collateral synapses in acute hippocampal slices with a chemical ischemia model. Baseline synaptic transmission was monitored for >10 min before ATP production was disrupted by switching to a modified extracellular solution lacking glucose and containing sodium azide and 2-deoxyglucose to inhibit oxidative phosphorylation and glycolysis, respectively (schematic in *Figure 1A*). This protocol for 'chemical ischemia' has previously been shown to reliably induce a rapid and reversible decline of the intracellular ATP levels (*Pape and Rose, 2023*). We first tested the effect of acute and transient chemical ischemia lasting for 2–5 min and observed two qualitatively distinct results. A comparison of recordings with 2 and 5 min illustrates both scenarios (*Figure 1B–F*). Although the postsynaptic response transiently disappeared for both durations, they remained suppressed over the next 50 min of recording for 5 min of chemical ischemia but recovered when chemical ischemia lasted only 2 min, in which case it showed a significant potentiation at the end of the recording (*Figure 1B, C, and E*). In contrast, the axonal fiber volley, which reflects the axonal firing of action potentials after stimulation, was transiently suppressed and recovered fully for both durations (*Figure 1D*). Analyzing the entire data set including chemical stress durations of 3 and 4 min confirmed that the axonal fiber volley robustly recovered from chemical ischemia for the tested durations (*Figure 1G*). In contrast, the postsynaptic response was either potentiated for shorter durations or almost completely suppressed for longer durations (*Figure 1H*). For chemical ischemia with near-complete suppression of the postsynaptic response (*Figure 1H*, orange data points), the average duration of chemical stress was 4.46±0.207 min (*n*=11) and 2.74±0.20 min (*n*=19) for all other recordings. Because during chemical ischemia the cellular depolarization and ensuing increased neuronal synaptic glutamate release should lead to an extracellular potassium increase, we measured extracellular [K$^+$] in parallel in a subset of experiments (example in *Figure 1F*). Correlating the maximum increase of the extracellular [K$^+$] during chemical ischemia with the lasting effect on synaptic transmission revealed that a stronger extracellular

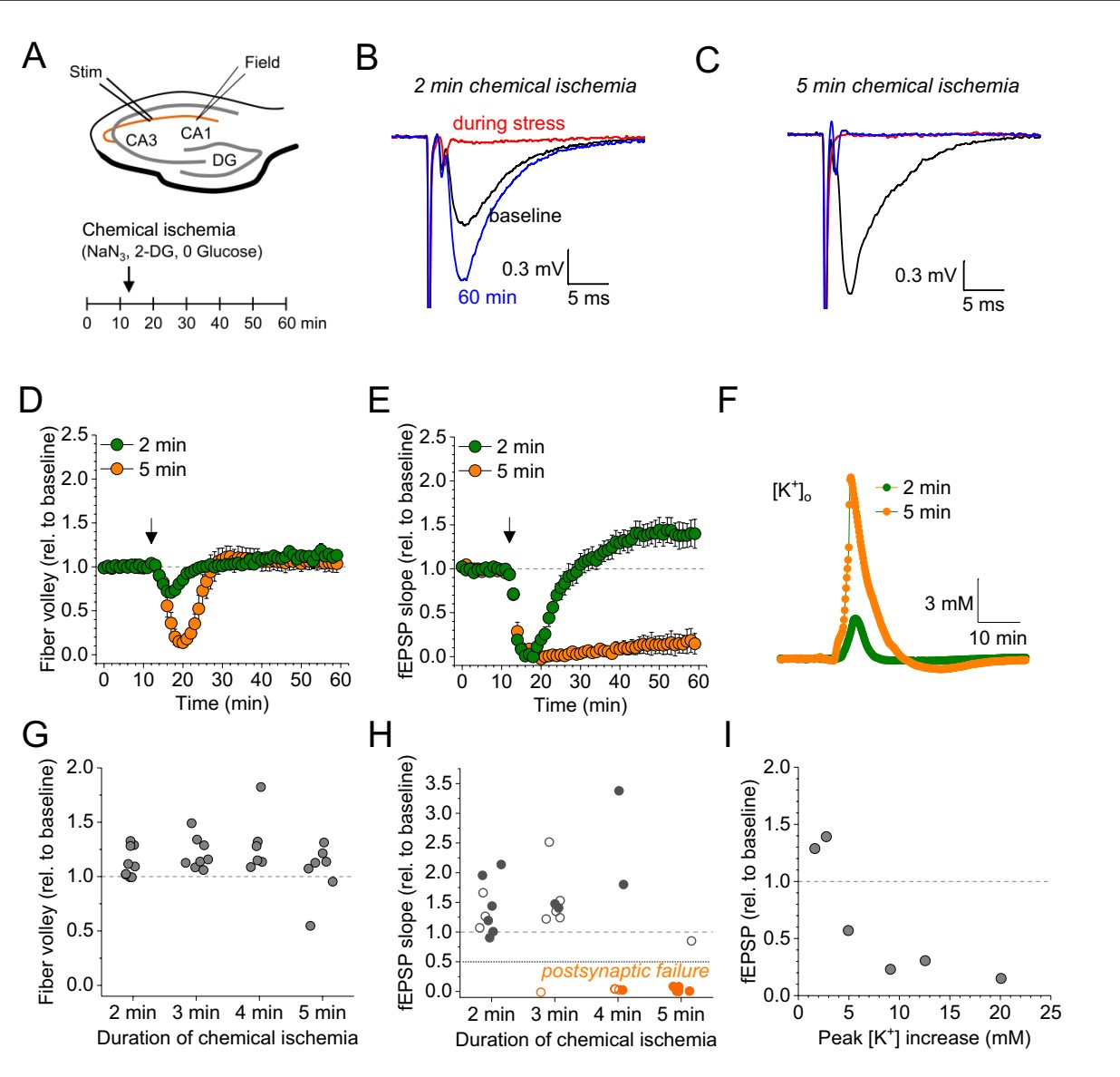

**Figure 1.** Duration-dependent and bidirectional effect of transient chemical ischemia on synaptic transmission. (**A**) Schematic of experimental design. Extracellular field potentials (Field) were recorded in the CA1 region in response to Schaffer collateral stimulation (Stim; paired pulses, 50 ms interstimulus interval, every 20 s). Arrow indicates the time point of application of a modified artificial cerebral spinal fluid (ACSF), inducing acute chemical ischemia, with 0 mM glucose, 2 mM deoxyglucose (2-DG), 5 mM sodium azide (NaN₃) for 2–5 min. (**B–C**) Example traces for 2 min chemical ischemia (**B**) and 5 min chemical ischemia (**C**) (black, baseline; red, during chemical ischemia; blue, end of the experiment at 60 min). (**D**) Relative change of the axonal fiber volley amplitude compared to baseline (0–10 min) for 2 min of chemical ischemia (green) and 5 min of chemical ischemia (orange). Arrow indicates time point of application of the modified ACSF for chemical ischemia induction. (**E**) Same as in (**D**) but for the field excitatory postsynaptic potential (fEPSP) slope. (**F**) Example traces of extracellular [K⁺] recordings during 2 and 5 min of chemical ischemia. (**G**) Quantification of the relative change of the axonal fiber volley amplitude in the last 10 min of recordings (50–60 min) relative to baseline for 2, 3, 4, and 5 min ($n$=9, 8, 6, and 7, respectively) of chemical ischemia. (**H**) Same experiments as in (**G**) but analysis of the fEPSP slope. Graphs in H and G include results from recording shown in **B–E**. Persistent failure of synaptic transmission highlighted in orange. Filled circles correspond to recordings, in which astrocytes expressed iGluSnFR (see **Figure 2** and text). (**I**) Change of fEPSP in the last 10 min of recordings (50–60 min) relative to the baseline plotted against maximum [K⁺] increase during chemical ischemia. Data are expressed as mean ± s.e.m.

[K⁺] increase (i.e. a severe metabolic failure) made postsynaptic failure more likely (**Figure 1I**). Together, these experiments reveal two distinct outcomes of acute chemical ischemia: either potentiation of the synaptic response or its near-complete suppression, which represents a postsynaptic failure of synaptic transmission as revealed by the following glutamate imaging experiments.

In a subset of the recordings shown in *Figure 1H* (filled data points), we have combined electrophysiological recordings with monitoring of glutamate release using the fluorescent indicator iGluSnFR (*Marvin et al., 2013*; *Figure 2*). In this subset, iGluSnFR was virally expressed in astrocytes and its fluorescence intensity changes were monitored in response to paired stimulation of CA3-CA1 synapses by line scanning throughout the experiment as indicated in *Figure 2A and B*. Additional control experiments confirmed the stability of such combined recordings (*Figure 2C and F*). We grouped recordings combining iGluSnFR imaging and electrophysiology according to the effect of chemical ischemia on the synaptic response: 'chemical ischemia with postsynaptic failure' (*Figure 1H*, filled orange data points) if the postsynaptic response did not recover to above 50% of the baseline level and 'chemical ischemia' when it did (*Figure 1H*, filled dark gray data points). Similar to electrophysiological recordings, two response types could be identified in iGluSnFR recordings. When acute and transient chemical ischemia led to a persistent potentiation of synaptic transmission, this was associated with an increase of the iGluSnFR responses ($\Delta F/F_0$) (*Figure 2D*). In contrast, if chemical ischemia led to postsynaptic failure, we observed a transient surge of the resting fluorescence of iGluSnFR ($F_0$) during acute chemical ischemia and reduced and gradually recovering iGluSnFR transients in response to synaptic stimulation after chemical ischemia (*Figure 2E*). Presented are six out of the nine experiments with postsynaptic failure that were performed. Three recordings were not further analyzed because the iGluSnFR fluorescence became undetectable after chemical ischemia. A more comprehensive statistical analysis of the responses before and after chemical ischemia can be found in *Figure 2—figure supplement 1* and *Figure 2F, G, and H*. In a single experiment, we have been able to record the response to two subsequent periods of short chemical ischemia in a >3 hr recording (*Figure 2—figure supplement 2*), which highlights the persistence of the potentiation of both the postsynaptic responses and glutamate transients. In addition to the bidirectional changes of synaptic transmission and glutamate release depending on the severity of chemical ischemia, we also found a consistent decrease of the iGluSnFR resting fluorescence over the duration of the experiments in all conditions. In general, this could be caused by a gradual reduction of the resting extracellular glutamate levels and/or a reduction of fluorescent iGluSnFR, for instance due to bleaching, which appears to be a likely explanation given the repeated imaging over up to 90 min. It is important to note that the increase of the iGluSnFR transients ($\Delta F/F_0$) associated with potentiated postsynaptic responses is unlikely due to the reduction of $F_0$ because the latter is observed in all experimental conditions. Similarly, it is unlikely that the increase of the iGluSnFR transients is because of the increased axonal excitability (increased fiber volley amplitude) because this was also observed in control recordings, where no change of the iGluSnFR response ($\Delta F/F_0$) was detected. The increase of the iGluSnFR response after synaptic stimulation also points to an increase of extracellular glutamate transients, because the iGluSnFR response to saturating glutamate concentrations is unaffected in experiments with chemical ischemia without postsynaptic failure (*Figure 2—figure supplement 3*).

Taken together these experiments reveal that longer durations of chemical ischemia result in a persistent postsynaptic failure with slowly recovering glutamate release. A likely reason for the postsynaptic failure is the death or persistent depolarization of CA1 pyramidal cells. Although not quantified, we indeed observed irreversible swelling and/or loss of integrity of CA1 pyramidal cell bodies in parallel to postsynaptic failure. This is in line with previous studies using transient metabolic inhibition with similar durations, which reported irreversible ATP depletion in neocortical neurons after transient chemical ischemia (*Pape and Rose, 2023*) and persistent depolarization of CA1 pyramidal cells after hypoxia combined with glucose deprivation (*Tanaka et al., 1997*). The beginning of this postsynaptic failure coincides with a surge of extracellular glutamate (*Figure 2E*), which we attribute to the increase of synaptic glutamate release at the onset of metabolic stress observed before (*Hershkowitz et al., 1993*; *Tanaka et al., 1997*), and of extracellular potassium (*Figure 1F and I*). Both are likely driven by neuronal depolarization, $Ca^{2+}$ accumulation, and the various other contributing mechanisms (*Passlick et al., 2021*). In contrast, shorter durations of chemical ischemia durations lead to a transient failure of synaptic transmission, possibly due to a transient depolarization of neurons, followed by an unexpected potentiation of synaptically driven glutamate transients and of postsynaptic responses.

## Glutamate uptake is unaffected after acute chemical ischemia

We next asked what the underlying reason for the increase glutamate transients is. One potential explanation is a restriction of extracellular space (ECS): If a given amount of glutamate is released

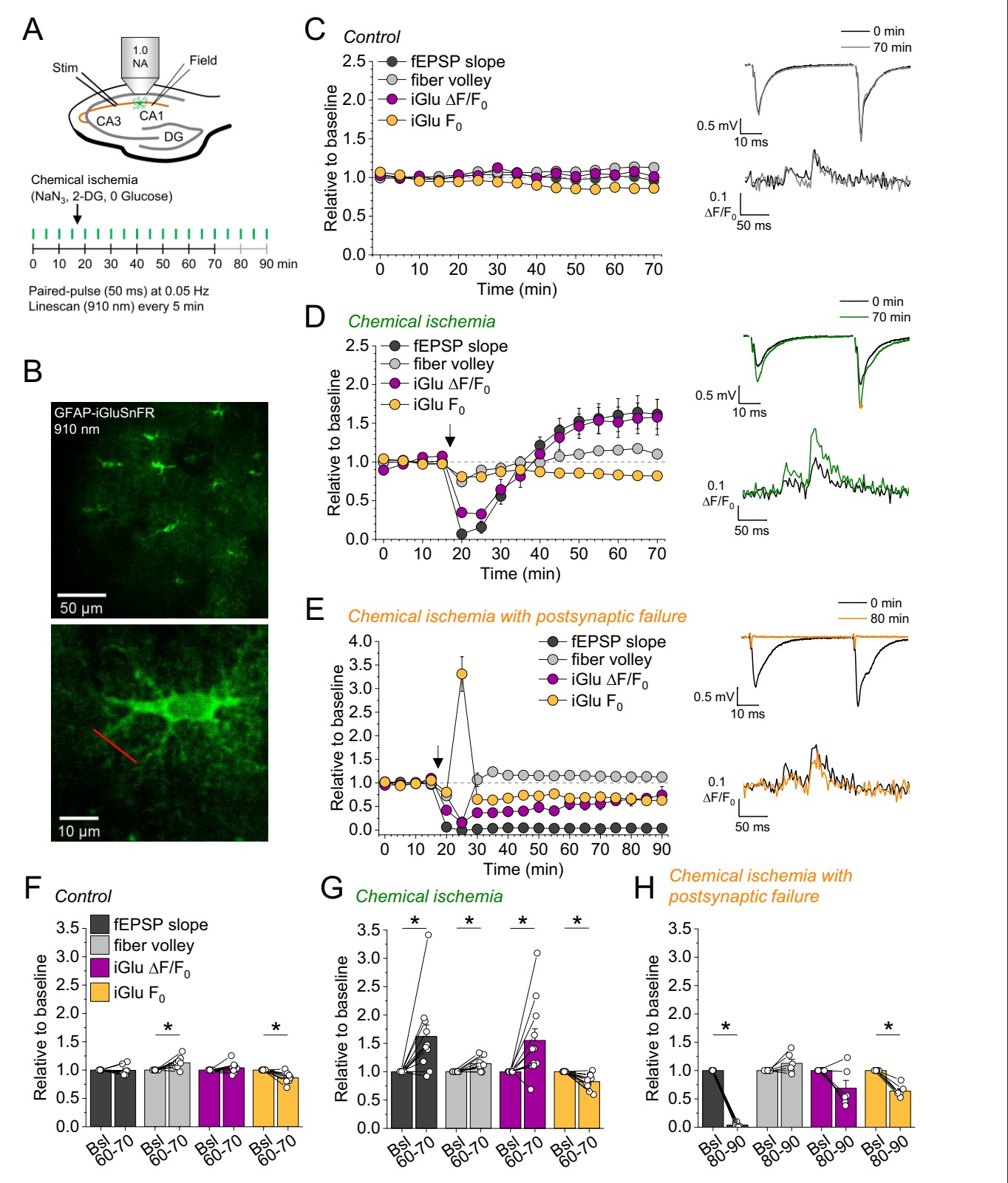

**Figure 2.** Bidirectional dysregulation of glutamate release by transient chemical ischemia. (**A**) Schematic of experimental design. Combined recording of extracellular field potentials (Field) in the CA1 region in response to Schaffer collateral stimulation (Stim; paired pulses, interstimulus interval of 50 ms, every 20 s) and two-photon excitation fluorescence line scan imaging (910 nm, 6× every 5 min) of the glutamate indicator iGluSnFR. (**B**) Top: example of virally induced iGluSnFR expression by astrocytes in the *stratum radiatum* of the CA1 region. Bottom: single iGluSnFR-expressing astrocyte and

*Figure 2 continued on next page*

*Figure 2 continued*

representative location of line scan (red line). (**C–E**) Left, field excitatory postsynaptic potential (fEPSP) slope, axonal fiber volley, iGluSnFR (iGlu) $\Delta F/F_0$ and iGlu resting fluorescence (F0) relative to baseline (0–10 min) for control (*n*=8) (**C**), chemical ischemia (*n*=11) (**D**), and chemical ischemia followed by postsynaptic failure of synaptic transmission (*n*=6) (**E**). Arrow indicates start of chemical ischemia. Right, example traces of fEPSP (top) and iGluSnFR $\Delta F/F_0$ (bottom, average of 6 scans) at the beginning and the end of the recording. The electrophysiological results in (**D** and **E**) are a subset from *Figure 1*. (**F–H**) Summary of parameters analyzed in (**C–E**) in the last 10 min of recording (60–70/80–90 min) compared to baseline (Bsl). (**F**) Control: fEPSP slope, p=0.894; fiber volley, p=0.012; iGlu $\Delta F/F_0$, p=0.310; iGlu $F_0$, p=0.006; *n*=8, paired Student's t-test. (**G**) Chemical ischemia: fEPSP slope, p=0.013; fiber volley, p=0.003; iGlu $\Delta F/F_0$, p=0.024; iGlu $F_0$, p=0.001; *n*=11, paired Student's t-test. (**H**) Chemical ischemia with postsynaptic failure: fEPSP slope, p<0.0001; fiber volley, p=0.134; iGlu $\Delta F/F_0$, p=0.073; iGlu $F_0$, p=0.0004; *n*=6, paired Student's t-test. Data are expressed as mean ± s.e.m.

The online version of this article includes the following figure supplement(s) for figure 2:

**Figure supplement 1.** Stimulus-response relationship of field excitatory postsynaptic potential (fEPSP) slope and iGluSnFR $\Delta F/F_0$ before and after chemical ischemia.

**Figure supplement 2.** Repeated chemical ischemia.

**Figure supplement 3.** Dynamic range of iGluSnFR is not affected by chemical ischemia without postsynaptic failure.

**Figure supplement 4.** The configuration of the extracellular space (ECS) is only affected after severe chemical ischemia.

into the ECS leading to a corresponding glutamate concentration, then a reduction of the ECS would increase the glutamate concentration. Indeed, anoxia and ischemia consistently lead to a reduction of the ECS fraction (*Syková and Nicholson, 2008*). This scenario was investigated using the method of combining tetramethylammonium (TMA) iontophoresis and TMA-selective microelectrode recordings for estimating the ECS fraction and extracellular diffusion (*Nicholson, 1993*; *Hrabětová and Nicholson, 2007*; *Syková and Nicholson, 2008*) in parallel to chemical ischemia induction. We found that chemical ischemia with postsynaptic failure was accompanied by a decreased ECS fraction and a decreased effective diffusion coefficient, which is consistent with previous reports (*Syková and Nicholson, 2008*; *Figure 2—figure supplement 4*). However, this was not observed for chemical ischemia without postsynaptic failure (*Figure 2—figure supplement 4*), in which the increased glutamate transients were observed.

An alternative explanation for the increased glutamate transients is reduced glutamate uptake. To investigate this possibility, we analyzed the decay time constant of the iGluSnFR fluorescence transients after synaptic stimulation, because it is increased when glutamate uptake is decreased (*Armbruster et al., 2016*; *Romanos et al., 2019*). However, we did not detect a significant change of the iGluSnFR decay time constant after chemical ischemia (*Figure 3A and B*). In addition, we tested if a deficit of glutamate uptake could be unmasked by increasing the amount of synaptically released glutamate. This was tested by high-frequency stimulation of CA3-CA1 synapses in the presence of glutamate receptor blockers (to prevent induction of synaptic plasticity, *Figure 3C*). Again, no differences of the decay time constant were observed between control recordings and recordings with chemical ischemia (*Figure 3E*). Finally, we verified the sensitivity of our experimental approach by inhibiting glutamate uptake pharmacologically at the end of the experiments. This manipulation indeed increased the decay time constant by a factor of ~10–20 (*Figure 3D, F, and G*). We therefore conclude that glutamate uptake is surprisingly resilient in the chemical ischemia model and that a reduction of glutamate uptake does not explain the increased extracellular glutamate transients after chemical ischemia without postsynaptic failure.

## Increased synaptic glutamate release after chemical ischemia

Our results point toward an increased synaptic glutamate release when chemical ischemia is shorter and followed by postsynaptic potentiation but not when it is associated with postsynaptic failure. Investigating the paired-pulse behavior of synaptic responses is a common approach for probing changes of presynaptic glutamate release (*Debanne et al., 1996*). Analyzing the paired-pulse ratio (PPR) of postsynaptic response and iGluSnFR transients revealed no consistent changes after chemical ischemia (*Figure 4—figure supplement 1*). However, even manifold presynaptically induced increases of synaptic strength were recently shown to result in relatively small changes of the PPR, especially of the iGluSnFR signal at single CA3-CA1 synapses (*Dürst et al., 2022*). This suggests that a more moderate presynaptic increase of glutamate release may escape its detection using the PPR. Therefore, we used an electrophysiological approach that directly detects changes of the glutamate concentration in the synaptic cleft. γ-D-glutamylglycine (γ-DGG) is a rapidly equilibrating competitive antagonist

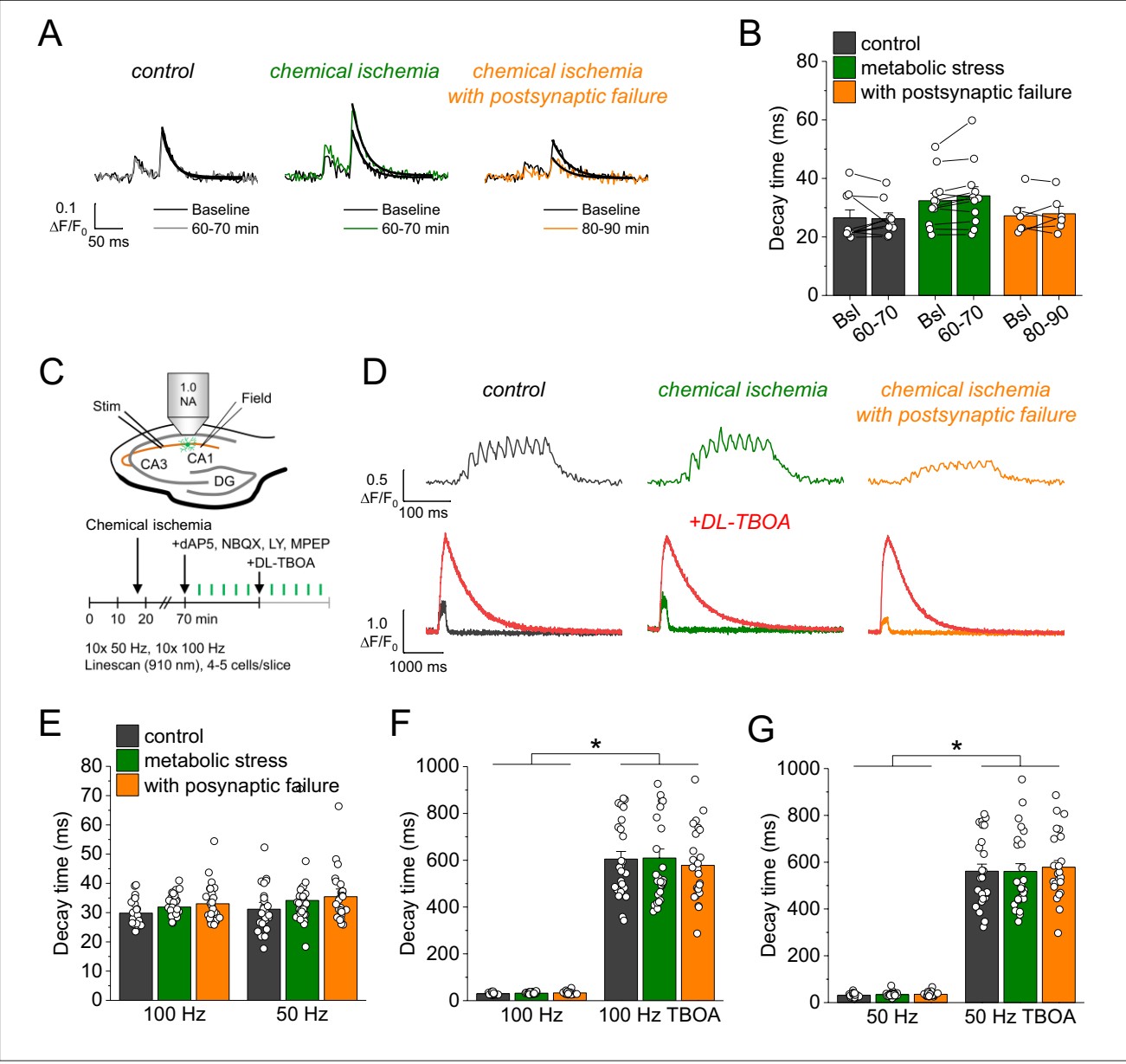

**Figure 3.** Glutamate clearance is not affected by transient chemical ischemia. (**A**) Example traces of iGluSnFR $\Delta F/F_0$ in response to paired-pulse stimulation (interstimulus interval 50 ms) during baseline (black) and last 10 min (60–70/80–90 min) of control (left, gray), chemical ischemia (middle, green), and chemical ischemia with postsynaptic failure (right, orange) recordings. Black line indicates exponential fit for the analysis of iGluSnFR decay time. (**B**) Quantification of iGluSnFR fluorescence decay time of the second pulse of the paired-pulse stimulation during the last 10 min of recording (60–70/80–90 min) compared to baseline (Bsl) for control, chemical ischemia, and chemical ischemia with postsynaptic failure. Control: p=0.860, *n*=9; chemical ischemia, p=0.062, *n*=12; chemical ischemia with postsynaptic failure, p=0.648, *n*=6; paired Student's t-tests. (**C**) Schematic of experimental design. Combined recording of extracellular field potentials (Field) in the CA1 region in response to Schaffer collateral stimulation (Stim; paired pulse [50 ms], 0.05 Hz) and two-photon excitation fluorescence line scan imaging (910 nm) of iGluSnFR. After 70 min, D-AP5 (50 μM), NBQX (20 μM), LY341495 (50 μM), and MPEP (10 μM) were added to the recording solution and iGluSnFR fluorescence changes ($\Delta F/F_0$) were recorded in 4–5 cells in response to 10× 50 Hz and 10× 100 Hz stimulation before and after application of DL-TBOA (100 μM). (**D**) Example traces of iGluSnFR $\Delta F/F_0$ line scan recordings (average of 6 scans) in response to 10× 50 Hz stimulation (top row) after control (left), chemical ischemia (middle), and chemical ischemia with postsynaptic failure recordings (right). The same cells were tested again after block of glutamate transporters by DL-TBOA (bottom row, red traces together with 'before' traces from upper row on different timescale for comparison). (**E**) Quantification of iGluSnFR fluorescence decay time in response to 10× 100 Hz and 10× 50 Hz stimulation after control, chemical ischemia recording without and with postsynaptic failure. 100 Hz: p=0.060, 50 Hz: p=0.149; *n*=25, 29, and 25 cells for control/chemical ischemia/chemical ischemia with postsynaptic failure from 5, 6, and 5 independent experiments, respectively; Kruskal-Wallis ANOVA. (**F**) Quantification of iGluSnFR fluorescence decay time in response to 10× 100 Hz stimulation in the presence and absence of DL-TBOA (same cells as in E). p<0.0001 for control, chemical ischemia, and chemical ischemia with postsynaptic failure, paired sample

*Figure 3 continued on next page*

*Figure 3 continued*

Wilcoxon signed-rank tests. (**G**) As in F but for 10× 50 Hz stimulation. p<0.0001 for control, chemical ischemia, and chemical ischemia with postsynaptic failure, paired sample Wilcoxon signed-rank tests. Data are expressed as mean ± s.e.m.

of AMPA receptors, and its inhibitory effect on AMPA receptor mediated postsynaptic responses depends on the ambient glutamate concentration because glutamate and γ-DGG compete for AMPA receptor binding (*Liu et al., 1999*; *Christie and Jahr, 2006*): The lower the glutamate concentration in the synaptic cleft, the stronger the inhibitory effect of γ-DGG, and vice versa. For performing this test after chemical ischemia without postsynaptic failure, we combined our previous experimental approach with whole-cell patch clamp recordings from CA1 pyramidal cells starting ~45 min after induction of chemical ischemia (*Figure 4A*). Analyzing the effect of γ-DGG application on simultaneously recorded field excitatory postsynaptic potentials (fEPSPs) and excitatory postsynaptic currents (EPSCs), we found that the percentages of fEPSP slopes and EPSC amplitudes remaining after γ-DGG

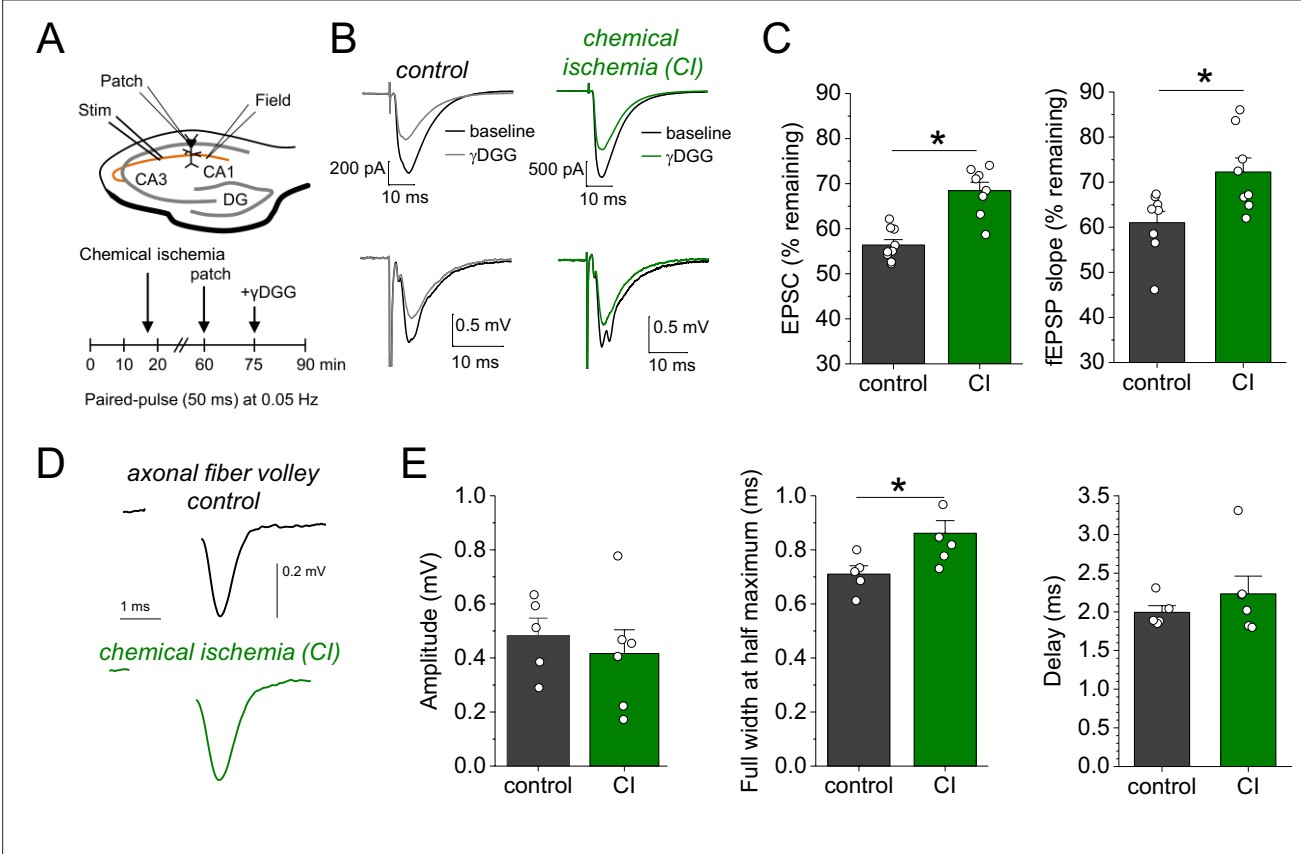

**Figure 4.** Chemical ischemia increases synaptic glutamate release. (**A**) Schematic of experimental design. Extracellular field potentials (Field) were recorded in the CA1 region in response to Schaffer collateral stimulation (Stim; paired pulse [50 ms], 0.05 Hz). After 60 min, a whole-cell patch clamp recording from a CA1 pyramidal neuron was started. After recording baseline excitatory postsynaptic current (EPSC) responses for 15 min, γ-D-glutamylglycine (γ-DGG) (1 mM) was added to the extracellular solution. (**B**) Example traces of EPSCs (top) and field excitatory postsynaptic potentials (fEPSPs) (bottom) before (black) and after γ-DGG application for control (left, gray) and chemical ischemia (right, green) recordings. (**C**) Quantification of the remaining EPSC amplitude (left) and fEPSP slope (right) (in % of the baseline value) after γ-DGG application for control (*n*=9) and chemical ischemia (*n*=8) recordings. EPSC, p<0.0001; fEPSP, p=0.014; paired two-sample Student's t-test. Data are expressed as mean ± s.e.m. (**D**) Example of axonal fiber volleys from experiments shown in *Figure 3C–E*. Stimulus artifact removed for clarity. (**E**) Comparison of the fiber volley amplitude (left, p=0.57), fiber volley half width at half maximum (middle, p=0.026), and fiber volley delay (time between onset of the stimulus and fiber volley peak, right, p=0.39). *n*=5 (control) and 6 (chemical ischemia), unpaired Student's t-tests.

The online version of this article includes the following figure supplement(s) for figure 4:

**Figure supplement 1.** Effect of transient chemical ischemia on short-term synaptic plasticity.

**Figure supplement 2.** Reduced glutamate concentration in synaptic vesicles in conditions typical of chemical ischemia.

were significantly higher in experiments with chemical ischemia compared to control experiments (*Figure 4B and C*). This indicates that chemical ischemia led to an increased peak glutamate concentration in the synaptic cleft, which is consistent with an increased synaptic glutamate release. Thus, shorter durations of chemical ischemia led to a potentiation of presynaptic glutamate release and of the postsynaptic response.

We further explored the underlying reason for this increase of glutamate release by computational modeling. First, we tested the hypothesis that presynaptic conditions expected for metabolic failure could increase vesicle loading with glutamate. This was analysed using a previously established model for simulating presynaptic vesicle loading (*Kolen et al., 2023*). As shown in *Figure 4—figure supplement 2*, accumulation of glutamate in the synaptic vesicle under control conditions reaches a steady state of close to 3500 molecules. Cytosolic conditions consistent with metabolic failure (decreasing pH, increasing $[Na^+]$ and $[Cl^-]$, decreasing $[K^+]$) all lead to a smaller number of glutamate molecules in the vesicle. Similarly, a decrease in the extracellular pH and a reduced vesicular ATPase activity (due to reduced ATP levels) also cause a decrease in the luminal glutamate concentration. As a control, we increased the cytosolic pH from 7.2 to 7.4 and observed a significant rise in the number of glutamate molecules. However, we are not aware of any reports regarding a presynaptic alkalinization during ischemia or chemical ischemia. Overall, these simulations show that the conditions that are typically observed during ischemia or metabolic failure tend to lower the vesicular glutamate concentration and therefore cannot explain the observed increased glutamate release. An alternative explanation is an increased probability of presynaptic vesicular release, which could result from increased presynaptic $Ca^{2+}$ influx, for instance, due to a presynaptic broadening of action potentials. For testing this, we analyzed the axonal fiber volley recorded in previous experiments (*Figure 3C and D*), in which glutamate receptors were blocked and the fiber volley was therefore isolated (*Figure 4D*). We found that the amplitude of the fiber volley was not different between control recordings and recordings after chemical ischemia without postsynaptic failure (*Figure 4E*, left panel), which confirms previous conclusions (*Figure 2*). However, the fiber volley was significantly broadened by ~20% (*Figure 4E*, middle panel) without a significant change of its delay after stimulation (*Figure 4E*, right panel). This could indicate that short transient chemical ischemia broadens the presynaptic action potential, which increases presynaptic $Ca^{2+}$ influx and presynaptic glutamate release and, thereby, the postsynaptic response.

## Discussion

We modeled pathophysiological scenarios with transiently reduced ATP levels such as PIDs and spreading depression (*Lauritzen et al., 2011*) by a chemical ischemia protocol. In our analysis, we focused on the recovery from this acute and transient chemical ischemia and its outcome. Depending on the duration of chemical ischemia, two qualitatively different changes of glutamatergic synaptic transmission became apparent. After prolonged chemical ischemia up to 5 min, the postsynaptic responses did not recover in contrast to presynaptic action potential firing and presynaptic glutamate release, which, however, did not completely return to baseline levels. After shorter chemical ischemia, the postsynaptic responses and presynaptic glutamate release recovered and were potentiated at the end of the recording whereas the axonal fiber volley amplitude was not different from control recordings. Overall, these observations reveal a graded vulnerability of the involved compartments to transient metabolic failure: Least resilient were postsynaptic neurons, followed by presynaptic glutamate release and axonal action potential firing. In contrast, glutamate clearance, which is mostly due to astrocytic glutamate uptake (*Danbolt, 2001*), was mostly unaffected even after long durations of chemical ischemia. This is in line with the observation from the hippocampus that transient chemical ischemia triggers much smaller ATP decreases and $Na^+$ increases, a measure of dysregulation of ion homeostasis, in astrocytes compared to neurons (*Meyer et al., 2022*; *Pape and Rose, 2023*). This is because of the strong expression of postsynaptic neuronal *N*-methyl-D-aspartate receptors and TRPV4 (*Pape and Rose, 2023*). It is also noteworthy that presynaptic glutamate release recovered to some extent even if the postsynaptic neurons remained unresponsive, thereby masking the recovery of glutamate release in electrophysiological recordings. We have not systematically tested durations of chemical ischemia beyond 5 min, because it is safe to assume this would further slow down, and ultimately prevent, the recovery of glutamate release and eventually also impair axonal action potential firing (*Figure 2E*).

The persistent potentiation of glutamate release after short chemical ischemia was unexpected. It was detected in electrophysiological recording using γ-DGG and iGluSnFR fluorescence imaging and it was not caused by a reduced glutamate uptake or changes of the ECS. Instead, a likely explanation is a broadening of presynaptic axonal action potentials, which is strongly suggested by the widened fiber volley in the absence of changes of fiber volley amplitude or delay. Nonetheless, a widened fiber volley could also be explained by, for instance, a desynchronization of presynaptic action potential firing. Presynaptic action potential broadening could explain increased glutamate release, because pharmacological prolongation of presynaptic action potentials by 4-AP was previously shown to increase postsynaptic responses by increasing presynaptic $Ca^{2+}$ influx at these synapses (*Wheeler et al., 1996*; *Qian and Saggau, 1999*). A potential reason for the action potential widening is an inactivation of potassium currents (*Mitterdorfer and Bean, 2002*), because of a lasting depolarization after transient chemical ischemia. Whether this is indeed the case remains to be tested experimentally. Alternatively, short durations of chemical ischemia could increase presynaptic glutamate release by persistently altering the presynaptic resting concentration of $Ca^{2+}$ (*Eshra et al., 2021*) or presynaptic cAMP signaling (*Vaden et al., 2019*), which are only two prominent alternative explanations among many others.

The time course of the potentiation of glutamate release and of the postsynaptic response were almost identical (*Figure 2D*), which suggests that the increased presynaptic release directly leads to the potentiation of the postsynaptic response. It has previously been shown that the potentiation of synaptic transmission (LTP) in ischemia models (ischemic/anoxic LTP) and physiological LTP can occlude each other (*Hsu and Huang, 1997*; *Stein et al., 2015*), which indicates that they can share the mechanisms underlying the potentiation of synaptic transmission. For CA3-CA1 synapses, there is a long-standing debate to what extent and under which conditions postsynaptic mechanisms such as receptor insertion and presynaptic changes of release underlie LTP (*Padamsey and Emptage, 2014*). Our experiments suggest that the potentiation of postsynaptic responses is at least in part due to increased glutamate release, but they also do not rule out a mix of pre- and postsynaptic changes.

Glutamate uptake can be used as an indirect measure of the astrocytic metabolic status, because it is mostly mediated by astrocytes and driven by their transmembrane $Na^+$ gradient and membrane potential (*Danbolt, 2001*; *Rossi et al., 2007*; *Passlick et al., 2021*), which are maintained by the $Na^+/K^+$-ATPase. In experiments with prolonged chemical ischemia with postsynaptic failure, we observed preserved glutamate uptake in two-thirds of the experiments whereas in one-third iGluSnFR expressed by astrocytes disappeared after chemical ischemia. While we do not know the exact reason for the latter, astrocyte metabolism seems to be overall more resilient to transient ischemia than the postsynaptic neurons. In experiments without postsynaptic failure, no lasting reduction of glutamate uptake was detected. Neither observation rules out that glutamate uptake was transiently impaired during or immediately following chemical ischemia, because in this time period astrocytic $Na^+$ levels were shown to increase reversibly (*Gerkau et al., 2018*; *Pape and Rose, 2023*). Similarly, they do not rule out that astrocytes are a transient source of glutamate during chemical ischemia by, for instance, reverse transport and channel-mediated glutamate release (*Rossi et al., 2000*; *Rossi et al., 2007*; *Yang et al., 2019*; *Passlick et al., 2021*). Taken together, lasting effects of chemical ischemia on glutamate uptake were insignificant compared to changes of presynaptic glutamate release and synaptic transmission.

In summary, our experiments reveal a bidirectional dysregulation of glutamate signaling after transient metabolic failure. Especially the lasting potentiation of synaptic glutamate release after short periods of energy depletion could be an important new aspect in the disease context. If individual PIDs lead to a similar potentiation of glutamate release, the latter could promote excitotoxic damage in the peri-infarct tissue. However, our experiments using chemical ischemia in acute hippocampal slices do not fully reproduce the complex situation in the brain during stroke or PIDs. While our approach allowed us to investigate the effect of variable degrees of cellular ATP depletion (*Pape and Rose, 2023*) on glutamatergic synapse function and to dissect the relevant cellular and synaptic mechanisms in detail, it investigates cellular changes independent of, for instance, blood circulation and the constraints imposed by a fixed skull volume. Another variable that could be of interest is sex, because many aspects of glutamate signaling such as receptor and transporter expression can depend on sex and/or estrous cycle (*Giacometti and Barker, 2020*). In the present study, we have performed experiments exclusively on male mice, leaving the question of a sex dependence of our observations

unanswered. It will be interesting to understand if such factors modify the bidirectional dysregulation of glutamate signaling observed here.

## Materials and methods

### Animals

Experiments were performed using 2- to 4-month-old male C57BL/6N mice (Charles River) that were kept under 12 hr light/dark conditions and had ad libitum access to food and water. All animal procedures were conducted in accordance with the regulations of the European Commission Directive 2010/63/EU and all relevant national and institutional guidelines and requirements. Procedures were further approved by the Landesamt für Natur, Umwelt und Verbraucherschutz Nordrhein-Westfalen (LANUV, Recklinghausen, Germany) where required.

### Stereotactic injections

In order to express the glutamate sensor iGluSnFR (*Marvin et al., 2013*) on the surface of astrocytes, 4-week-old mice were injected bilaterally with an AAV expressing iGluSnFR under control of the GFAP promoter (AAV1.GFAP.iGluSnFr.WPRE.SV40, PennCore) into the CA1 region of the dorsal hippocampus. Mice were deeply anesthetized by intraperitoneal injection (i.p.) of fentanyl/midazolam/medetomidine (0.05/5.0/0.5 mg/kg bodyweight, injection volume 0.1 ml/10 g bodyweight). After deep anesthesia was confirmed, the head was shaved, the skin disinfected, and the head fixed in a stereotactic frame (Model 901, David Kopf Instruments). An incision was made, bregma localized, and a small hole was drilled with a dental drill (coordinates for the dorsal hippocampus, relative to bregma: anterior −1.8 mm, lateral ±1.6 mm, ventral −1.6 mm). Next, a beveled needle nanosyringe (nanofil 34 G BVLD, WPI) was slowly inserted into the brain and 0.5–1 µl of viral particles were injected into the hippocampus using a microinjection pump (50 nl/min, WPI). The injection needle was left in place for about 3 min and then slowly retracted. After repeating the procedure for the other hemisphere, the incision was sutured using absorbable thread (Ethicon) and an antibiotic was applied (Refobacin 1 mg/g, Gentamicin). Finally, anesthesia was terminated by i.p. injection of naloxone/flumazenil/atipamezole (1.2/0.5/2.5 mg/kg bodyweight, injection volume 0.1 ml/10 g bodyweight). Analgesia was applied 30 min before terminating the anesthesia and 24 hr after surgery by subcutaneous injection of carprofen (5 mg/kg bodyweight, injection volume 0.1 ml/20 g bodyweight). Animals were used for experiments 2–5 weeks after virus injection.

### Hippocampal slice preparation

Electrophysiology and two-photon excitation fluorescence microscopy in acute hippocampal slices were combined as previously described (*Minge et al., 2017*). Briefly, coronal slices were obtained from mice and virus-injected mice (see above) with a thickness of 300 µm. Slices were prepared in an ice-cold slicing solution containing (in mM): NaCl 60, sucrose 105, KCl 2.5, $MgCl_2$ 7, $NaH_2PO_4$ 1.25, ascorbic acid 1.3, sodium pyruvate 3, $NaHCO_3$ 26, $CaCl_2$ 0.5, and glucose 10 (osmolarity 300–310 mOsm/l), and kept in the slicing solution at 34°C for 15 min before being stored at room temperature (21–23°C) in an artificial cerebral spinal fluid (ACSF) containing (in mM): NaCl 131, KCl 2.5, $MgSO_4$ 1.3, $CaCl_2$ 2, $NaH_2PO_4$ 1.25, $NaHCO_3$ 21, and glucose 10 (osmolarity adjusted to 295–305 mOsm/l). Slices were allowed to rest for at least 60 min. For recordings, slices were transferred to a submersion-type recording chamber and perfused with extracellular solution (ACSF). All recordings were performed at 33–35°C. All solutions were continuously bubbled with 95% $O_2$/5% $CO_2$. The flow rate of the extracellular solution was kept constant at 6–7 ml/min throughout all experiments.

### Electrophysiology

Extracellular recordings of fEPSPs were performed using patch pipettes filled with extracellular solution placed in the stratum radiatum of the CA1 region. For stimulation of CA3-CA1 Schaffer collateral axonal connections, a bipolar concentric stimulation electrode was placed in the stratum radiatum at the border between CA2/3 and CA1. Stimulation intensities (DS3, Digitimer Ltd., UK) were set to obtain ~50% of the maximum fEPSP amplitude. The duration of individual stimuli was 100 µs. In some experiments, field recordings were combined with whole-cell patch clamp recordings from CA1 pyramidal cells using standard patch pipettes (3–4 MΩ) filled with an intracellular solution containing

(in mM): KCH$_3$O$_3$S 135, HEPES 10, di-tris-phosphocreatine 10, MgCl$_2$ 4, Na$_2$-ATP 4, Na-GTP 0.4, 0.2 BAPTA, 5 QX314-Cl (pH adjusted to 7.2, osmolarity 290–295 mOsm/l). CA1 pyramidal cells were identified using infrared differential interference contrast or Dodt contrast optics by their typical morphology and location. Data were recorded using MultiClamp 700B (Molecular Devices) amplifiers, digitized (10 kHz) and stored for offline analysis. Whole-cell patch clamp recordings were rejected if at any time the access resistance exceeded 25 MΩ or changed by more than 20%.

Acute chemical ischemia was induced by switching from normal ACSF to a modified ACSF with 0 mM glucose, 2 mM deoxyglucose (2-DG) to inhibit ATP production through glycolysis, and 5 mM sodium azide (NaN$_3$) to inhibit ATP production through oxidative phosphorylation. The modified ACSF was applied for variable durations ranging from 2 to 5 min. In some experiments, the inhibitors D-AP5 (dAP5 in figures, 50 µM, Abcam), NBQX disodium salt (10 µM, Abcam), DL-TBOA (100 µM, Tocris), LY341495 (50 µM, Tocris), MPEP (10 µM, Tocris), and/or γ-DGG (1 mM, Tocris) were added to the extracellular solution as indicated.

## Glutamate imaging using iGluSnFR

Glutamate imaging using iGluSnFR (*Marvin et al., 2013*) in combination with electrophysiology was performed similarly as previously described (*Herde et al., 2020*). Slices with astrocytes expressing iGluSnFR after viral injection (see above) were transferred to a submersion-type recording chamber mounted on a Scientifica two-photon excitation fluorescence microscope with a ×60/1.0 NA objective (Olympus). iGluSnFR was excited using a femtosecond pulse laser (Vision S, Coherent, excitation wavelength of 910 nm). During imaging, line scans of iGluSnFR fluorescence were repeatedly obtained. The scanned line was positioned in the periphery of an iGluSnFR-expressing astrocytes (e.g. *Figure 2B*). The line was scanned at a frequency of 378.8 Hz to obtain a fluorescence profile over time. Line scans were repeated four to six times for each individual time point and experiment and averaged before the background fluorescence was subtracted to obtain an average iGluSnFR fluorescence time course. The resting iGluSnFR fluorescence before synaptic stimulation ($F_0$), the maximum increase of iGluSnFR fluorescence ($\Delta F$), and the normalized iGluSnFR time course ($\Delta F/F_0$) were determined. iGluSnFR imaging was combined with paired synaptic stimuli (*Figure 2*), high-frequency synaptic stimulation (*Figure 3*), and iontophoretic glutamate application (*Figure 2—figure supplement 2*). For the latter, an iontophoretic application system (MVCS-02C-150, NPI) was used with iontophoretic pipettes (60–80 MΩ resistance) filled with 150 mM glutamic acid (pH adjusted to 7.0 with NaOH) and 40 µM Alexa Fluor 633 to localize the pipette.

## Measurements of ECS fraction and diffusivity and extracellular K$^+$

Extracellular [K$^+$] was recorded using K$^+$-sensitive microelectrodes as previously described (*Breithausen et al., 2020*). Briefly, these microelectrodes were built using theta-glass capillaries and pulled with a horizontal pipette puller. The reference and the K$^+$-sensitive barrel were filled with an NaCl (154 mM) and KCl (150 mM) solution, respectively. The tip of the K$^+$-sensitive barrel was filled with a valinomycin-based K$^+$ ionophore (K$^+$ ionophore 1, cocktail B, Sigma-Aldrich). The K$^+$-sensitive microelectrodes were calibrated before each experiment with solutions containing 154 mM NaCl and 3 or 30 mM KCl. Electrodes were used if they responded to this 10× increase of K$^+$ with a voltage response >50 mV.

Measurements of the relative changes of the ECS fraction and of extracellular diffusion during experiments with acute chemical ischemia were obtained using the method of combining TMA iontophoresis and TMA-sensitive electrode recordings (*Nicholson, 1993*; *Hrabětová and Nicholson, 2007*; *Syková and Nicholson, 2008*). TMA-sensitive, double-barrel microelectrodes were built like K$^+$-sensitive microelectrodes except that the TMA-sensitive ionophore solution (IE190, WPI) and 150 mM TMA chloride, as the intrapipette solution, were used for the TMA-sensitive barrel. The reference barrel was filled with 154 mM NaCl and 60 µM Alexa Fluor 594 to visualize the electrode. Recordings were performed using an ION-01M amplifier (NPI, Germany). TMA-sensitive microelectrodes were tested for their sensitivity to TMA before each experiment by two calibration solutions containing either (in mM) TMA 1, KCl 3, NaCl 150 or TMA 10, KCl 3, NaCl 141. Only electrodes that responded to this 10-fold increase of TMA with a potential increase of >50 mV were used for the experiments. A subset of electrodes was also tested for their sensitivity over a larger range of TMA concentrations, which revealed Nernst-like behavior over higher concentrations and only a departure

from that in the low sub-millimolar range. To correct for that, the calibration curve was fitted with the Nicolsky-Eisenman equation and the conversion of the TMA-electrode potential into a TMA concentration was adapted accordingly (*Nicholson, 1993*). For recordings in chemical ischemia experiments (*Figure 2—figure supplement 4*), the TMA-sensitive microelectrode was placed into the CA1 stratum radiatum and an iontophoresis pipette filled with 100 mM TMA and 60 µM Alexa Fluor 594 was placed at distance of 100–200 µm from the TMA-sensitive microelectrode. At each time point during the recording, TMA was iontophoretically injected into the tissue (100 nA, 30 s) and the corresponding [TMA] time course was calculated. For example, see *Figure 2—figure supplement 4*. Each time course was fitted using equations 10.2–10.4 from *Hrabětová and Nicholson, 2007*, using a MATLAB script to obtain the ECS fraction and the effective TMA diffusion coefficient (D*). D* deviates from the free diffusion coefficient of TMA (D) because the tortuosity of the ECS reduces TMA diffusion over the distances investigated here (*Hrabětová and Nicholson, 2007*). Therefore, changes of D* can indicate a change of ECS tortuosity. In these experiments, the extracellular solution always contained 1 mM TMACl to provide a known basal extracellular TMA concentration for the conversion of the recorded microelectrode potential into [TMA]. The reference barrel of the TMA-sensitive microelectrode was also used for recording fEPSPs in response to axonal stimulation. We have verified these recordings to report accurate values by recording [TMA] transients in agarose gel (ECS fraction ~1, tortuosity ~1) (*Nicholson, 1993*) and in acute brain slices, in which we obtained an ECS fraction of 0.19±0.023 (*n*=5) and a tortuosity of 1.54±0.046 (*n*=5) and which matches previous measurements in this region (*Nicholson, 1993*; *Syková and Nicholson, 2008*).

## Glutamate accumulation in synaptic vesicles

The accumulation of glutamate in synaptic vesicles is modeled as previously described (*Kolen et al., 2023*) and the code for the model is archived along with it. Briefly, the temporal evolution of luminal glutamate molecule number, [H⁺], pH, and [Cl⁻] are modeled by the following rate equations:

$$\frac{d\left[Glut\right]_L}{dt} = J_{VGLUT}, \tag{1}$$

$$\frac{d\left[H^+\right]_L}{dt} = J_{H^+} + J_{V_{ATPase}} - J_{VGLUT}, \tag{2}$$

$$\frac{dpH_L}{dt} = -\frac{1}{N_A}\left(\frac{dH_L^+}{dt}\frac{1}{V}\right)\left(\frac{1}{\beta_{pH}}\right), \tag{3}$$

$$\frac{d\left[Cl^-\right]_L}{dt} = J_{VGLUT_{Cl}}. \tag{4}$$

where $N_A$, $\beta_{pH}$ ($=\frac{40\text{mM}}{pH}$), and $V$ represent Avogadro's number, the pH-buffering capacity of the vesicle, and the vesicle volume assuming a radius of 20 nm, respectively. The equations for glutamate/H⁺ flux through glutamate transporters ($J_{VGLUT}$), Cl⁻ flux through glutamate transporters ($J_{VGLUT_{Cl}}$), proton leak ($J_{H^+}$), and V-type ATPase ($J_{V_{ATPase}}$) are described in *Kolen et al., 2023*.

The equation for the membrane potential of the vesicle ($\Delta\psi$) is

$$\Delta\psi = \frac{F \times V}{C}\left(\left[H^+\right]_L + \left[K^+\right]_L + \left[Na^+\right]_L - \left[Cl^-\right]_L - \left[Glut\right]_L - B\right) \tag{5}$$

with $C = C_0 \times S$ being the total capacitance of the vesicle with $C_0$=1 µF/cm², $S$ the surface area of the vesicle, and $F$ Faraday's constant. Luminal K⁺ ($[K^+]_L$) and Na⁺ ($[Na^+]_L$) concentrations were fixed to extracellular values, i.e., to 5 and 145 mM, respectively. $B$ is the luminal concentration of impermeant charges given by the conservation of charge in the vesicle under initial conditions (indicated by the subscript 0).

$$B = \left[H^+\right]_{L,0} + \left[K^+\right]_{L,0} + \left[Na^+\right]_{L,0} - \left[Cl^-\right]_{L,0} - \left[Glut\right]_{L,0} - \frac{C}{F \times V}\left(\psi_{in} - \psi_{out}\right) \tag{6}$$

Finally, the effect of surface charge on various ion concentrations at the membrane was also taken into account, with the inner ($\psi_{in}$) and outer ($\psi_{out}$) leaflets potentials set to 0 and –50 mV, respectively.

Starting conditions for the mathematical modeling reflect the ion concentrations immediately after endocytosis of synaptic vesicles: the vesicular lumen contained solutions resembling the external

solutions, with high [Cl$^-$], neutral pH, and negligible glutamate concentration ($[K^+]_{L,0}$=5 mM, $[Na^+]_{L,0}$ = 145 mM, $[Cl^-]_{L,0}$ = 110 mM, $[Glut]_{L,0}$ = 1 μM, and $pH_{L,0}$ = 7.2); cytoplasmic pH, Cl$^-$, K$^+$, Na$^+$, and glutamate concentration was set to 7.2, 10 mM, 140 mM, 10 mM, and 10 mM, respectively. We also incorporated changes in luminal Na$^+$ and K$^+$ concentrations, however, both did not affect our results.

## Statistics

Analyses were performed using ImageJ (NIH), Origin (OriginLab), and MATLAB (MathWorks). Numerical data are reported as mean ± s.e.m. with *n* being the number of samples. In all electrophysiological and imaging experiments in acute hippocampal slices, *n* refers to the number of recordings. Because the success rate of experiments combining electrophysiology, two-photon excitation imaging, and pharmacology is low, a single successful experiment was typically performed per day and animal. For all other experimental designs, detailed information can be found in the figure legends. The Shapiro-Wilk test was used to establish if data were normally distributed. Comparisons were then performed using the appropriate parametric and non-parametric tests (e.g. Student's t-test or Mann-Whitney U-test). The statistical tests used are indicated throughout. In figures, asterisks indicate statistical significance. Exact p values are provided in the figure legends. Error bars represent s.e.m. and dots represent individual data points, which are connected by lines for paired data points.

## Acknowledgements

We thank Gerald Seifert and Thomas Erdmann for support with animal breeding and maintenance and Temitope Adeoye for useful discussions on synaptic vesicle release. The study was supported by the German Research Foundation (DFG, FOR2795, to CH) and the National Institutes of Health (Grant number R01NS130916, to GU).

## Additional information

### Funding

| Funder | Grant reference number | Author |
| --- | --- | --- |
| Deutsche Forschungsgemeinschaft | FOR2795 | Christian Henneberger |
| National Institutes of Health | R01NS130916 | Ghanim Ullah |
| German Research Foundation | FOR2795 | Christian Henneberger |
| National Institutes of Health | R01NS130916 | Ghanim Ullah |

The funders had no role in study design, data collection and interpretation, or the decision to submit the work for publication.

### Author contributions

Stefan Passlick, Conceptualization, Data curation, Visualization, Methodology, Writing – review and editing; Ghanim Ullah, Data curation, Investigation, Visualization, Writing – review and editing; Christian Henneberger, Conceptualization, Resources, Supervision, Funding acquisition, Investigation, Visualization, Methodology, Writing – original draft, Project administration, Writing – review and editing

### Author ORCIDs

Stefan Passlick ⓘ https://orcid.org/0000-0002-7703-7018
Christian Henneberger ⓘ https://orcid.org/0000-0002-5391-7387

### Ethics

All animal procedures were conducted in accordance with the regulations of the European Commission Directive 2010/63/EU and all relevant national and institutional guidelines and requirements.

Procedures were further approved by the Landesamt für Natur, Umwelt und Verbraucherschutz Nordrhein-Westfalen (LANUV, Recklinghausen, Germany) where required.

Reviewer #1 (Public Review): https://doi.org/10.7554/eLife.98834.3.sa1
Reviewer #2 (Public Review): https://doi.org/10.7554/eLife.98834.3.sa2
Reviewer #3 (Public Review): https://doi.org/10.7554/eLife.98834.3.sa3
Author response https://doi.org/10.7554/eLife.98834.3.sa4

---

## Additional files

### Supplementary files
• MDAR checklist

### Data availability

All data needed to evaluate the conclusions in the paper are presented in the paper and/or the supplements. Source data for main results are provided via DRYAD (DOI: https://doi.org/10.5061/dryad.qjq2bvqr1).

The following dataset was generated:

| Author(s) | Year | Dataset title | Dataset URL | Database and Identifier |
| --- | --- | --- | --- | --- |
| Passlick S, Ullah G, Henneberger C | 2024 | Data from: Bidirectional dysregulation of synaptic glutamate signaling after transient metabolic failure | https://doi.org/10.5061/dryad.qjq2bvqr1 | Dryad Digital Repository, 10.5061/dryad.qjq2bvqr1 |

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
