## [Editor Report · eLife assessment]

The authors show that short bouts of chemical ischemia lead to presynaptic changes in glutamate release and long-term potentiation, whereas longer bouts of chemical ischemia lead to synaptic failure and presumably cell death. This **convincing** work relies on rigorous electrophysiology/imaging experiments and data analysis. It is **important** as it provides new mechanistic details on chemical ischemia, which could offer potential insights into ischemic stroke in vivo.

---

## [Referee Report · Reviewer #1 (Public Review)]

Summary:

This work by Passlick and colleagues set out to reveal the mechanism by which short bouts of ischemia perturb glutamate signalling. This manuscript builds upon previous work in the field that reported a paradoxical increase in synaptic transmission following acute, transient ischemia termed ischemic or anoxic long-term potentiation. Despite these observations how this occurs and the involvement of glutamate release and uptake mechanisms remained unanswered.

Here the authors employed two distinct chemical ischemia models, one lasting 2-minutes, the other 5-minutes. Recording evoked field excitatory postsynaptic potentials in acute brain slices, the authors revealed that shorter bouts of ischemia resulted in a transient decrease in postsynaptic responses followed by an overshoot and long-term potentiation. Longer bouts of chemical ischemia (5-minutes), however, resulted in synaptic failure that did not return to baseline levels over 50-minutes of recording (Figure 1).

Two-photon Imaging of fluorescent glutamate sensor iGluSnFR expressed in astrocytes matched postsynaptic responses with shorter ischemia resulting in a transient dip before increase in extracellular glutamate which was not the case with prolonged ischemia (Figure 2).

Mechanistically, the authors show that this increased glutamate levels and postsynaptic responses were not due to changes in glutamate clearance (Figure 3). Next using a competitive antagonist for postsynaptic AMPA receptors the authors show that synaptic glutamate release was enhanced by 2-minute chemical ischemia.

Taken together, these data reveal the underlying mechanism regarding ischemic long-term potentiation, highlighting presynaptic release as the primary culprit. Additionally, the authors show relative insensitivity of glutamate uptake mechanisms during ischemia, highlighting the resilience of astrocytes to this metabolic challenge.

---

## [Referee Report · Reviewer #2 (Public Review)]

Summary:

To investigate the impact of chemical ischemia induced by blocking mitochondrial function and glycolysis, the authors measured extracellular field potentials, performed whole-cell patch-clamp recordings, and measured glutamate release with optical techniques. They found that shorter two-minutes-lasting blockade of energy production initially blocked synaptic transmission but subsequently caused a potentiation of synaptic transmission due to increased glutamate release. In contrast, longer five-minutes-lasting blockage of energy production caused a sustained decrease of synaptic transmission. A correlation between the increase of extracellular potassium concentration and the response upon chemical ischemia indicates that the severity of the ischemia determines whether synapses potentiate or depress upon chemical ischemia. A subsequent mechanistic analysis revealed that the speed of uptake of glutamate is unchanged. An increase in the duration of the fiber volley reflecting the extracellular voltage of the action potentials of the axon bundle was interpreted as an action potential broadening, which could provide mechanistic explanation. In summary, the data convincingly demonstrate that synaptic potentiation induced by chemical ischemia is caused by increased glutamate release.

Strengths:

The manuscript is well written, and the experiments are carefully designed. The results are exciting, novel, and important for the field. The main strength of the manuscript is the combination of electrophysiological recordings and optical glutamate imaging. The main conclusion of increased glutamate release was furthermore supported with an independent approach relying on a low-affinity competitive antagonist of glutamate receptors. The data are of exceptional quality. Several important controls were carefully performed, such as the stability of the recordings and the size of the extracellular space. The number of experiments are sufficient for the conclusions. The careful data analysis justifies the classification of two types of responses, namely synaptic potentiation and depression after chemical ischemia. The data are carefully discussed and the conclusions are justified.

Weaknesses:

The weaknesses are minor. The authors measured the fiber volley, which reflects the extracellular voltage of the compound action potential of the fiber bundle. The half-duration of the fiber volley was increased. These results are consistent with action potential broadening in the axons but the action potential broadening was not experimentally demonstrated. However, these results are carefully discussed.

---

## [Referee Report · Reviewer #3 (Public Review)]

Summary:

This valuable study shows that shorter episodes (2min duration) of energy depletion, as it occurs in ischemia, could lead to long lasting dysregulation of synaptic transmission with presynaptic alterations of glutamate release at the CA3-CA1 synapses. A longer duration of chemical ischemia (5 min) permanently suppresses synaptic transmission. By using electrophysiological approaches, including field and patch clamp recordings, combined to imaging studies, the authors demonstrated that 2 min of chemical ischemia leads to a prolonged potentiation of synaptic activity with a long lasting increase of glutamate release from presynaptic terminals. This was observed as an increase in iGluSnFR fluorescence, a sensor for glutamate expressed selectively on hippocampal astrocytes by viral injection. The increase in iGluSnFR fluorescence upon 2 min chemical ischemia could not be ascribed to an altered glutamate uptake, which is unaffected by both 2 min and 5 min chemical ischemia. The presynaptic increase in glutamate release upon short episodes of chemical ischemia is confirmed by a reduced inhibitory effect of the competitive antagonist gamma-D-glutamylglycine on AMPA receptor mediated postsynaptic responses. Fiber volley durations in field recording are prolonged in slices exposed to 2 min chemical ischemia. The authors interpret this data as an indication that the increase in glutamate release could be ascribed to a prolongation of the presynaptic action potential possibly due to inactivation of voltage-dependent K+ channels. However, more direct evidence are needed to fully support this hypothesis. This research highlights an important mechanism by which altered ionic homeostasis underlying metabolic failure can impact on neuronal activity. Moreover, it also showed a different vulnerability of mechanisms involved in glutamatergic transmission with a marked resilience of glutamate uptake to chemical ischemia.

Strengths:

(1) The authors use a variety of experimental techniques ranging from electrophysiology to imaging to study the contribution of several mechanisms underlying the effect of chemical ischemia on synaptic transmission.

(2) The experiments are appropriately designed and clearly described in the figures and in the text.

(3) The controls are appropriate

Weaknesses:

- The results are obtained in an ex-vivo preparation

Impact:

This study provides a more comprehensive view of the long term effects of energy depletion during short episodes of experimental ischemia leading to the notion that not only post-synaptic changes, as reported by others, but also presynaptic changes are responsible for long-lasting modification of synaptic transmission. Interestingly, the direction of synaptic changes is bidirectional and dependent on the duration of chemical ischemia, indicating that different mechanisms involved in synaptic transmission are differently affected by energy depletion.

---

## [Author Response]

The following is the authors’ response to the original reviews.

**Public Reviews:**

**Reviewer #1 (Public Review):**
[…]Weaknesses:The question of the physiological relevance of short bouts of ischemia remains.

The chemical ischemia protocol induces a duration-dependent ATP depletion in acute slices on a time scale of minutes (Pape and Rose 2023). This is about the same time scale as the peri-infarct depolarisation (Lauritzen et al. 2011) that the protocol attempts to model. Of course, such models do not completely replicate the complex situation in vivo. However, the presented analyses of synapse function cannot be performed in vivo. We discuss this now in the manuscript.

The precise mechanisms underlying the shift between ischemia-induced long-term potentiation and long-term failure of synaptic responses were not addressed. Could this be cell death?

Thank you for the comment. Yes, we indeed believe that the persistent failure of synaptic transmission is because of neuronal cell death (i.e., of CA1 pyramidal cells) or at least persistent depolarisation. We did not explicitly state that in the original submission but do so in the revised manuscript. It is supported by the unquantified observation of swelling and/or loss of integrity of CA1 pyramidal cell bodies in parallel to postsynaptic failure. It is also in line with many reports from the literature, of which we now cite two (lines 186-198).

Sex differences are not addressed or considered.

We have performed all experiments on male mice, as indicated in Material and Methods. We have indeed not addressed sex differences of the observed effects. We consider this, and many other important factors, to be interesting topics for follow-up studies. This is now discussed (lines 413-424).

**Reviewer #2 (Public Review):**
[…]Weaknesses:The weaknesses are minor and only relate to the interpretation of some of the data regarding the presynaptic mechanisms causing the potentiation of release. The authors measured the fiber volley, which reflects the extracellular voltage of the compound action potential of the fiber bundle. The half-duration of the fiber volley was increased, which could be due to the action potential broadening of the individual axons but could also be due to differences in conduction velocity. We are therefore skeptical whether the conclusion of action broadening is justified.

These are excellent points. We have added an analysis demonstrating that axonal conduction velocity is unlikely to be affected. Nonetheless, the fiber volley is indeed an indirect measure of what happens in individual axons. We have adjusted our interpretation accordingly and now also discuss alternative explanations of our findings (lines 363-379).

**Reviewer #3 (Public Review):**
[…]Weaknesses:The data on fiber volley duration should be supported by more direct measurements to prove that chemical ischemia increases presynaptic Ca2+ influx due to a presynaptic broadening of action potentials. Given the influence that positioning of the stimulating and recording electrode can have on the fiber volley properties, I found this data insufficient to support the assumption of a relationship between increased iGluSnFR fluorescence, action potential broadening, and increased presynaptic Ca2+ levels.

We have added a new analysis showing that the latency of the fiber volley is unaffected and relatively constant, which strengthens our conclusion. But the fiber volley is indeed an indirect measure of action potential firing in individual axons. The suggested experiment, which would require simultaneous recording of Ca2+ and action potentials in single axons in combination with chemical ischemia, is extremely difficult, if possible at all. Instead, we have extended the discussion and include now further alternative mechanistic explanations (lines 363-379).

The results are obtained in an ex-vivo preparation, it would be interesting to assess if they could be replicated in vivo models of cerebral ischemia.

This would certainly be very interesting but also extremely challenging technically. For a detailed analysis of synaptic changes as presented here, the main difficulty will be to stimulate and visualise glutamate release exclusively in an isolated population of synapses while recording postsynaptic responses in a stroke model.

**Recommendations For The Authors:**

**Reviewer #1 (Recommendations For The Authors):**
[…]Labelling of experimental groups of 2-minute and 5-minute chemical ischemia is more accurate than "metabolic stress" and "with postsynaptic failure". The critical difference between these two conditions is lost with this nomenclature. The reader could be misled to believe that the two groups form a heterogenous population of responses from the same experimental manipulation which is incorrect.

We had stated in the manuscript that we ‘ … grouped combined iGluSnFR and electrophysiological recordings according to the effect of chemical ischemia on the synaptic response: ‘chemical ischemia with postsynaptic failure’ if the postsynaptic response did not recover to above 50% of the baseline level and ‘chemical ischemia’ when it did (as indicated in Fig. 1H). …’. The recordings were not grouped according to chemical stress duration but according to the effect on the postsynaptic response. We have revised the text explaining this (lines 125-135) and illustrate that now also in Fig. 1H. We hope this is easier to follow now.

More details on the long-term impact of 5-minute ischemia on cell viability would be enlightening regarding the specific mechanism separating these two conditions. With 2 minutes it would appear that cells remain alive (i.e. intact post-synaptic responses), 5 minutes however, inducing cell death.

Yes, our observations, although not quantified, are in line with cell death as CA1 pyramidal cell bodies appeared swollen and/or lost their integrity when chemical ischemia was followed by postsynaptic failure. This is also in line with reports from the literature. We have revised the results section accordingly (lines 186-201).

In the paragraph titled "glutamate uptake is unaffected after acute chemical ischemia", there are two erroneous citations of Figure S3 that should be Figure S4.

Thank you. We corrected this mistake.

The sex of animals is not given. This is essential information.

We used male mice as indicated in the initial version of the manuscript (Material and Methods). We have added a statement regarding the role of sex to the final section of the Discussion.

**Reviewer #2 (Recommendations For The Authors):**
We propose addressing the weaknesses mentioned in the public review. As said, the fibre volley is a very indirect measure of action potential broadening. Based on the iGluSnFR data, the authors predict that the potentiation is mediated by depolarization, action potential broadening, and increased presynaptic calcium influx. The latter could be tested experimentally, but this does not seem necessary if the data are interpreted more cautiously. For example, other explanations for the broadened fiber volley could be mentioned, such as a slowing and/or dispersion of the action potential propagation speed. Furthermore, depolarization could cause elevated resting calcium concentrations, which could potentiate release independently of action potential broadening. Finally, classical forms of presynaptic potentiation of the release machinery that occur during homeostatic plasticity or Hebbian plasticity may operate independently of calcium dynamics.

Thank you for this comment. The discussion of the mechanism was indeed too short. We have added an analysis of the fiber volley delay after stimulation, which was not affected. Presynaptic action potential broadening is, in our opinion, a very likely explanation for our observations but we did not perform direct experiments. Directly recording presynaptic action potentials and Ca2+ transients in the chemical ischemia model over extended periods of time is a major technical challenge and certainly of interest in the future. As suggested, we have expanded the discussion section and now mention various alternative explanations (lines 363-379).

There are the following minor suggestions:Add line numbers.

We have added line numbers.

We would suggest providing exact P values instead of asterisks in the figures.

We agree that having exact P values in the figure panels can be very helpful. However, in the present figures they are hard to integrate without overcrowding the already complex panels and thereby obscuring other important details. All p-values are included in the figure legends and/or main text.

Abstract: "We also observed an unexpected hierarchy of vulnerability of the involved mechanisms and cell types." This sentence is hard to understand and cell types were not directly compared (i.e. axons of CA3 and axons of CA1 neurons were not compared).

We have revised this statement and removed the reference to cell types.

In Figure 1G there seems to be an increase in the fiber volley. Is this significant? Could this be due to swelling of the slice during chemical ischemia? Or an increase in excitability? Maybe this could be discussed.

The effect was analysed in the context of Fig. 2. A significant increase of the fiber volley amplitude was detected in chemical ischemia (Fig. 2H) but also under control conditions (Fig. 2F). We therefore consider this a change that is detectable but not related to chemical ischemia and not a potential explanation for increased glutamate release (lines 157-160). Also, no significant fiber volley increase was detected in chemical ischemia with postsynaptic failure (Fig. 2H) and in the experiments illustrated in Fig. 4E. Our interpretation is that the fiber volley unspecifically increases in some experiments over the time course of the experiment (~ 60 min) but this is unrelated to chemical ischemia.

In the results: "A fully separate set of experiments..." Please explain better what this means.

We have revised the entire section to explain more clearly how recordings were grouped (lines 125135).

In the results: "...(Syková and Nicholson, 2008) (Figure S3). However, this was not observed for chemical ischemia without postsynaptic failure (Figure S3), in which the increased glutamate transients were observed." This should probably refer to Figure S4.

Thank you for spotting this mistake. We corrected it.

The last sentence in the results "... most likely by increased presynaptic Ca2+ influx, and, at the same time, the postsynaptic response." This is difficult to understand. Does "at the same time" refer to another mechanism or the consequence of more Ca2+?

We revised this part of the results section to improve clarity and toned down our conclusions (lines 328-335 and 363-379).

**Reviewer #3 (Recommendations For The Authors):**
There are a few points that the author needs to clarify:The authors do not discuss the different behaviour of iGlu F0 during chemical ischemia and chemical ischemia with postsynaptic failure shown in Figure 2, panels D and E. In the first case, during the application of the solution to induce ischemia, iGluF0 decreases while in the other case, it strongly increases before falling down. In both cases, the fEPSP slope is decreased. How does the author explain this observation?

We attribute the transient increase of extracellular glutamate during prolonged chemical ischemia to the increase of synaptic glutamate release observed previously under such conditions (Hershkowitz et al. 1993; Tanaka et al. 1997) and other mechanisms reviewed by us (Passlick et al. 2021) (e.g., glial glutamate release, transiently reduced glutamate uptake), which we could not detect during shorter chemical ischemia. The initial drop of the fEPSP slope is most likely due to postsynaptic depolarisation, which is followed by a repolarisation if the chemical stress duration is short. We now explain this in more detail in lines 185-200 of the revised manuscript. Although we focussed on the bi-directional effect on longer timescales in this manuscript, this transient phase during chemical ischemia is very interesting for further investigations.

On page 8, first line, I think that the authors meant Figure S4, not Figure S3 when they mentioned results on ECS diffusivity and ECS fraction.

Yes, thank you for spotting this. We corrected the mistake.

In Supplementary Figure 5 panel B It seems that PPR is significantly reduced upon chemical ischemia (asterisk on columns green) but the authors claimed in the paper at page 10 that "Analysing the paired-pulse ratio (PPR) of postsynaptic response and iGluSnFR transients revealed no consistent changes after chemical ischemia (Figure S5).". Did the authors refer to the data normalized in panel D? In this case, I do not see the need to normalize raw data that have been already shown in a previous panel and that give different statistical results, probably due to the different tests used (paired in panel B and not paired in panel D).

We have clarified this point in the supplementary material (Figure S5, legend). There is a relevant difference between the analyses presented in panel B and D. The paired test presented in B analyses the change of the electrophysiological PPR in response to chemical ischemia. The test in D on the electrophysiologically PPR asks if the reduction in B is significantly different from the changes seen under control conditions. Because it is not, we conclude that chemical ischemia has no relevant effect on the electrophysiological PPR and, in combination with the results on the iGluSnFR PPR, also not on short-term plasticity, as tested here.

References

Hershkowitz N, Katchman AN, Veregge S. Site of synaptic depression during hypoxia: a patch-clamp analysis. Journal of Neurophysiology 69: 432–441, 1993.

Lauritzen M, Dreier JP, Fabricius M, Hartings JA, Graf R, Strong AJ. Clinical Relevance of Cortical Spreading Depression in Neurological Disorders: Migraine, Malignant Stroke, Subarachnoid and Intracranial Hemorrhage, and Traumatic Brain Injury. J Cereb Blood Flow Metab 31: 17–35, 2011.

Pape N, Rose CR. Activation of TRPV4 channels promotes the loss of cellular ATP in organotypic slices of the mouse neocortex exposed to chemical ischemia. The Journal of Physiology 601: 2975–2990, 2023.

Passlick S, Rose CR, Petzold GC, Henneberger C. Disruption of Glutamate Transport and Homeostasis by Acute Metabolic Stress. Front Cell Neurosci 15: 637784, 2021.

Tanaka E, Yamamoto S, Kudo Y, Mihara S, Higashi H. Mechanisms Underlying the Rapid

Depolarization Produced by Deprivation of Oxygen and Glucose in Rat Hippocampal CA1 Neurons In Vitro. Journal of Neurophysiology 78: 891–902, 1997.